# Pre-Harvest Salicylic Acid Application Affects Fruit Quality and Yield under Deficit Irrigation in *Aristotelia chilensis* (Mol.) Plants

**DOI:** 10.3390/plants12183279

**Published:** 2023-09-15

**Authors:** Jorge González-Villagra, León A. Bravo, Marjorie Reyes-Díaz, Jerry D. Cohen, Alejandra Ribera-Fonseca, Rafael López-Olivari, Emilio Jorquera-Fontena, Ricardo Tighe-Neira

**Affiliations:** 1Departamento de Ciencias Agropecuarias y Acuícolas, Facultad de Recursos Naturales, Universidad Católica de Temuco, Temuco P.O. Box 15-D, Chile; ejorquera@uct.cl (E.J.-F.); rtighe@uct.cl (R.T.-N.); 2Núcleo de Investigación en Producción Alimentaria, Facultad de Recursos Naturales, Universidad Católica de Temuco, Temuco P.O. Box 15-D, Chile; 3Departamento de Ciencias Agronómicas y Recursos Naturales, Facultad de Ciencias Agropecuarias y Medioambiente, Universidad de La Frontera, Temuco P.O. Box 54-D, Chile; leon.bravo@ufrontera.cl; 4Center of Plant, Soil Interaction and Natural Resources Biotechnology, Scientific and Technological Bioresource Nucleus (BIOREN), Universidad de La Frontera, Temuco P.O. Box 54-D, Chile; marjorie.reyes@ufrontera.cl (M.R.-D.); alejandra.ribera@ufrontera.cl (A.R.-F.); 5Departamento de Ciencias Químicas y Recursos Naturales, Facultad de Ingeniería y Ciencias, Universidad de La Frontera, Temuco P.O. Box 54-D, Chile; 6Department of Horticultural Science, University of Minnesota, St. Paul, MN 55108, USA; cohen047@umn.edu; 7Centro de Fruticultura, Facultad de Ciencias Agropecuarias y Medioambiente, Campus Andrés Bello, Universidad de La Frontera, Temuco P.O. Box 54-D, Chile; 8Instituto de Investigaciones Agropecuarias, INIA Carillanca, Km 10 camino Cajón-Vilcún s/n, Temuco P.O. Box 929, Chile; rafael.lopez@inia.cl

**Keywords:** plant water status, fruit growth, equatorial diameter, soluble solids, total phenols

## Abstract

Salicylic acid (SA) application is a promising agronomic tool. However, studies under field conditions are required, to confirm the potential benefits of SA. Thus, SA application was evaluated under field conditions for its effect on abscisic acid levels, antioxidant related-parameters, fruit quality, and yield in *Aristotelia chilensis* subjected to different levels of irrigation. During two growing seasons, three-year-old plants under field conditions were subjected to full irrigation (FI: 100% of reference evapotranspiration (ETo), and deficit irrigation (DI: 60% ETo). During each growth season, a single application of 0.5 mM SA was performed at fruit color change by spraying fruits and leaves of both irrigation treatments. The results showed that DI plants experienced moderate water stress (−1.3 MPa), which increased ABA levels and oxidative stress in the leaves. The SA application facilitated the recovery of all physiological parameters under the DI condition, increasing fruit fresh weight by 44%, with a 27% increase in fruit dry weight, a 1 mm increase in equatorial diameter, a 27% improvement in yield per plant and a 27% increase in total yield, with lesser oxidative stress and tissue ABA levels in leaves. Also, SA application significantly increased (by about 10%) the values of fruit trait variables such as soluble solids, total phenols, and antioxidant activity, with the exceptions of titratable acidity and total anthocyanins, which did not vary. The results demonstrated that SA application might be used as an agronomic strategy to improve fruit yield and quality, representing a saving of 40% regarding water use.

## 1. Introduction

Water deficit is considered one of the most important environmental stress factors for crop plants, reducing yields by more than 50%, causing a severe threat to food security [1,2,3]. It has been estimated that by 2025 50% of the Earth’s surface will consist of lands under water deficit [4]. Chile is considered to likely be one of the most water-deficit areas of the world by 2040, and the precipitation deficit in Central–South Chile (30° to 41° S) has already reached between 40 and 75% during the last decade, the so-called “Mega Drought” [5,6,7,8]. Reductions in annual precipitation will increase the water deficit phenomenon, resulting in less water for crop production, and risking significant reductions in crop yields. Thus, the development of water saving practices is crucial in order to maintain or increase crop yields while using less water [9,10]. Deficit irrigation (DI) is an agronomic practice to reduce crop water consumption, with no or only minimal reduction in yield [11,12]. However, contradictory results have been reported in deficit irrigation studies, where Ertek and Kara [13] showed that a 30% reduction in the irrigation supply had no impacts on crop yield, and values of 45% lower water reduced yields by about 15% compared to full irrigation in field-grown *Zea mays*. Similarly, Silveira et al. [14] reported no significant changes in fruit yield in *Citrus sinensis* plants under different DI treatments (25, 50, and 75% DI). In contrast, Tari [15] found that between 35 and 65% of DI increased water use efficiency (WUE) in *Triticum aestivum*; however, crop yields fell by about 22% compared to full irrigation when irrigation was applied at the stem elongation and heading stages. In *Prunus avium*, deficit irrigation applied at the third phase of fruit development slightly reduced plant yield, with no apparent impact on fruit quality and with a positive effect on leaf WUE [16]. In *Vaccinum corymbosum*, cutting off irrigation late in fruit development reduced yield, leading to smaller, but the firmest, berries produced in comparison with full irrigation [17]. Similar results in berry quality were reported by Ribera-Fonseca et al. [18], who applied 40% less irrigation water compared to the full irrigation control in pot-grown *V. corymbosum* plants. Therefore, the effect of DI on fruit quality and yield seems to depend on multiple factors, including the irrigation level, the developmental stage in which water deficit is applied, the species, and/or other experimental conditions.

In coordination with the annual growth cycle, plant hormones regulate plant responses to offset the deleterious effects of different environmental stresses, including a reduction in water availability [19,20]. Central to a plant’s response to decreased water, abscisic acid (ABA) modulates drought stress by changes in stomatal aperture as well as mechanisms such as enzymatic and non-enzymatic antioxidants, decreasing oxidative stress [19,21,22]. In addition, other, more recently recognized as stress responsive, phytohormones play important roles, and these include brassinosteroids, jasmonates, melatonin, strigolactones, and salicylic acid [23,24]. Among these signaling molecules, salicylic acid (SA) has been recognized as a key phytohormone involved in plant responses to environmental and biological stress by regulating physiological processes such as photosynthesis and nutrient uptake, and by inducing the expression of the antioxidant defense system, primarily in studies under controlled and greenhouse conditions [25,26,27]. Recently, Wakchaure et al. [28] observed that 20 µM SA treatments improved *Abelmoschus esculentus* yield and fruit quality subjected to different deficit irrigation conditions. However, additional studies are needed to understand SA’s possible role under water deficit in horticultural crops under field conditions [29].

*Aristotelia chilensis* (Mol.) Stuntz (Elaeocarpaceae), commonly known as “maqui”, is an important berry growing in Southern Chile [30,31]. Maqui fruits are widely recognized for their high antioxidant capacity and phenolic compound levels, with human health properties such as cardioprotective, anti-inflammatory, and antidiabetic activities, thus becoming to an important industrial crop [32,33,34]. In addition, maqui fruits have experienced an important increase in the export market during recent years, especially to several countries including Japan, South Korea, Italy, the United States, and Germany [35,36]. Currently, the high demand for this species has led to several studies related to crop management, morphological and physiological characterization to improve its domestication and agronomic practices [31,37,38]. In a previous study, we [39] reported that foliar 0.5 mM SA application improved plant water status, photosynthesis, and plant growth in *A. chilensis* plants subjected to 40% of deficit irrigation compared to non-SA-treated plants under greenhouse conditions. Interestingly, our results showed that SA triggered the antioxidant defense mechanism, including higher activities of superoxide dismutase (SOD) and ascorbate peroxidase (APX), and higher levels of phenolic compounds in *A. chilensis* (Mol.) Stuntz plants subjected to deficit irrigation. However, the effects of SA application on yield and fruit quality have been less well-documented in fruit crops, and no prior studies address *A. chilensis* plants and stress responses while growing under field conditions.

Thus, the aim of this study was to provide the data needed by determining the response to SA application on ABA, antioxidant-related parameters, fruit quality, and yield in *A. chilensis* plants subjected to deficit irrigation under field conditions.

## 2. Results

### 2.1. Environmental Conditions, Applied Water, and Plant Water Status

The maximum temperature (T_max_) typically ranged between 25 and 26 °C during the first season, where, with a few exceptions, days reached 30 °C during January at the end of the experiment (Figure 1A). In the second season there was a slight increase, with T_max_ ranging between 28 and 30 °C, mainly during December (Figure 1C). The seasonal ETo values were 271 and 299 mm for the 2020/2021 and 2021/2022 seasons, respectively (Figure 1B,D). Otherwise, rainfall reached 115 mm for the 2020/2021 season, and 80 mm for the 2021/2022 season (Figure 1B,D and Table 1). Cumulative applied water is shown in Table 1, where irrigation was around 2700 m^3^ ha^−1^ for the 2021/2022 season, and 2900 m^3^ ha^−1^ for the 2021/2022 season. With respect to stem water potential (Ψ_w_), plants subjected to DI showed lower levels (around −1.2 MPa) compared to FI plants (around −0.6 MPa) during both seasons (Table 2). Interestingly, SA application significantly improved the Ψ_w_ in DI plants, reducing to around 15% of Ψ_w_ compared to non-SA-treated plants, showing the same behavior during both evaluated seasons (Table 2).

### 2.2. ABA Levels in Leaves

Plants subjected to deficit irrigation (DI) exhibited greater ABA levels compared to FI treatment in our experiment (Figure 2). The increments were 38 and 17% for DI plants during the 2020/2021 and 2021/2022 seasons, respectively. Thus, SA application decreased ABA levels in *A. chilensis* plants subjected to DI treatment. ABA level was decreased by 20% for DI plants compared to non-SA-treated DI plants during the first season (2020/2021). In comparison, in DI plants subjected to SA application, ABA levels were reduced by 13% compared to DI plants without SA application (Figure 2). Also, SA application significantly decreased ABA levels in plants under FI condition compared to plants without SA application, during both seasons.

### 2.3. Lipid Peroxidation in Leaves

Lipid peroxidation (LP) was measured as an oxidative stress indicator in *A. chilensis* plants. Deficit irrigation (DI) exhibited significantly higher LP values of around 12 and 28%, compared to full irrigation (FI) treatment during the 2020/2021 and 2021/2022 seasons, respectively (Figure 3). When SA was applied to *A. chilensis* plants subjected to DI, LP was significantly reduced in both seasons. LP levels were reduced by 40 and 42% in DI SA-treated plants compared to DI non-SA-treated plants for 2020/2021 and 2021/2022 seasons, respectively (Figure 3). Interestingly, SA application also reduced LP levels in plants subjected to FI treatment compared to FI treatment without SA application during both seasons.

### 2.4. Yield and Fruit Quality

The DI treatment negatively affected fruit growth and yield in *A. chilensis* plants during both seasons (Table 3). Thus, fruit fresh weight, fruit dry weight, and yield per plant were reduced by about 30% in response to DI treatment in contrast to FI treatment during both seasons. In contrast, SA application recovered fruit fresh weight, fruit dry weight, yield per plant, and total yield parameters, mainly during the second season (44%, 27%, 1 mm, 27%, and 27%, respectively) in DI plants. However, DI treatment decreased by 13% and 20% the equatorial diameter of fruits (for similar values of fruit growth and yield) during the 2020/2021 and 2021/2022 seasons, respectively (Table 4). Interestingly, SA application improved equatorial dimeter (by about 1 mm) in DI plants compared to non-SA-treated plants subjected to DI treatment during both seasons. In contrast, polar diameter of fruits was not affected by any treatment in *A. chilensis* plants. On the other hand, total soluble solids (TSS) were increased in DI treatment, being 32% and 22% higher compared to FI treatment during the 2020/2021 and 2021/2022 seasons, respectively. No significant variations in TA were observed in *A. chilensis* fruits in our experiments during both seasons.

### 2.5. Antioxidant Parameters in Fruits

In order to provide a more comprehensive analysis of the fruit quality related to antioxidant properties of *A. chilensis* fruits, antioxidant activity (AA), total phenols (TP), and total anthocyanins (ANT) were determined. The study showed that AA slightly increased (18.5%) in plants subjected to DI treatment compared to FI treatment during the 2020/2021 season (Figure 4A). DI plants increased their AA levels by 11% relative to plants subjected to FI treatment in the 2021/2022 season. Interestingly, SA application to DI plants improved AA levels by about 20% in contrast to FI treatment during both seasons. A similar tendency was observed in total phenols (TP), where DI plants showed slightly higher levels (12%) compared to FI in the 2020/2021 season (Figure 4B). No changes were observed in TP levels between DI plants and FI plants during the 2021/2022 season. However, SA application resulted in greater TP levels in plants under DI conditions compared to non-SA-treated plants, being 21% higher mainly during the 2021/2022 season. Total anthocyanins (ANT) increased by 21% and 29% in DI plants during the 2020/2021 and 2021/2022 seasons, respectively (Figure 4C). Although SA application increased ANT, no additional changes were noted in ANT in plants subjected to DI conditions.

## 3. Discussion

Deficit irrigation (DI) is rising as an important agronomic practice for reducing water consumption in the context of the water deficit that affects several agricultural areas worldwide [11,12]. Also, salicylic acid (SA) application has become a promising agronomic tool for improving plant growth and yield under water deficit [28,39]. However, more studies are needed under field conditions to establish these methods as practical solutions for agricultural practice.

In our study, the water applied in the full irrigation treatment (FI) was around 2700 and 2900 m^3^ ha^−1^ for the 2021/2022 and 2021/2022 seasons, respectively. Notably, 40% less water was applied under DI in both seasons (Table 1). For Ψ_S_, FI plants showed an average value of −0.6 MPa, while DI plants showed an average value of −1.3 MPa for both seasons, which was increased by 15% with SA application to DI plants (Table 2). According to our previous study, Ψ_S_ of −1.3 MPa could be considered as a moderate water stress in *A. chilensis* plants because lipid peroxidation increased by about 45% and stomatal conductance (g_s_) was reduced by between 40 and 60% for DI with respect to FI [39,40], similar to that reported in *Vitis vinifera*, where a 50%-to-60% reduction in g_s_ values is considered as moderate water stress [41].

It is widely reported that plants subjected to water stress show increases in abscisic acid (ABA), which regulates both stomatal closure and defense mechanism such as enzymatic and non-enzymatic antioxidants to decrease lipid peroxidation [19,22]. We found that deficit-irrigation plants exhibited greater ABA levels (38% for 2020/2021 and 17% for 2021/2022) compared to FI plants (Figure 2). However, SA application decreased ABA levels by 20% in DI treatment for 2020/2021 and 13% for 2021/2022, compared to non-SA-treated DI treatment (Figure 2). Similar results were recently reported by Iqbal et al. [42], where SA shown an antagonistic interaction with ABA, resulting in reduced ABA biosynthesis and increased plant biomass in *Brassica juncea* plants subjected to water deficit. Interestingly, La et al. [43] reported that *Brassica napus* plants growing under water deficit with SA application exhibited lower ABA levels compared to plants under similar soil water condition but without SA application. These authors also showed that SA application significantly decreased hydrogen peroxide (H_2_O_2_) and the superoxide anion radical (O_2_^−^) in *B. napus* plants, which are important ROS, provoking lipid peroxidation and leading to cell injury and death. We observed that SA application significantly reduced lipid peroxidation (LP) levels in leaves of DI plants (Figure 3). Similar results were observed in our previous study, where 0.5 mM SA application reduced LP levels in *A. chilensis* plants subjected to water deficit in greenhouse conditions [39]. Furthermore, the reduction in oxidative stress was associated with higher levels of phenolic compounds and antioxidant activity induced by SA in leaves of *A. chilensis* plants, which improved carbon assimilation rate and plant growth.

The results revealed that fruit fresh weight, fruit dry weight, equatorial diameter, fruit production per plant, and yield were negatively affected by DI treatment during both seasons. However, SA application recovered all parameters in DI plants (mainly during the 2021/2022 season), increasing by 44% fruit fresh weight, 27% fruit dry weight, 1 mm equatorial diameter, 27% yield per plant, and total yield. Meanwhile, polar diameter of fruits was not affected by DI or SA application in *A. chilensis* plants. We found similar equatorial diameter in *A. chilensis* fruits as reported by Fredes et al. [44], who collected fruits from wild plants. Our results also agree with Giménez et al. [45] and García-Pastor et al. [46], who reported that SA application increased fruit growth and yield in *P. avium* and *Punica granatum* plants, in a dose-dependent way. Giménez et al. [45] showed that 0.5 mM increased by 41% fruit growth in *P. avium* cultivars Sweetheart and Sweet Late, with respect to non-SA-treated plants. Iqbal et al. [42] observed that SA inhibited stomatal closure mediated by ABA under water deficit, increasing photosynthesis and growth. In fact, we previously observed under controlled conditions that stomatal conductance, CO_2_ assimilation and growth were higher in *A. chilensis* plants subjected to water deficit with SA application in greenhouses [39]. Therefore, we can suggest that SA could improve fruit growth and yield, decreasing oxidative stress and ABA levels in *A. chilensis* plants subjected to water deficit. It is important to highlight that we found higher fruit quality and yield parameters (by about 15%) compared to *A. chilensis* plants irrigated at 100% of evapotranspiration, as reported in Yañez et al. [31] and González et al. [47]. The authors analyzed fruits of plants from different geographical provenances but established in the same experimental garden (Maule Region, Chile), which showed the high genetic diversity in *A. chilensis*. TSS in fruits with DI treatment increased by 32% and 22% compared to FI treatment during the 2020/2021 and 2021/2022 seasons, respectively, and unchanged TSS levels were observed in fruit of DI plants with SA application. Meanwhile, TA remained unchanged between DI and FI treatments, with no significant variations under SA application during both seasons (Table 4). Fruits of *A. chilensis* plants are widely recognized for their greater levels of phenolic compounds and antioxidant capacity [32,34]. In our study, SA application improved AA levels in DI plants by about 20% in contrast to FI treatment, during both seasons. Similarly, SA application exhibited greater TP levels in DI plants compared to non-SA-treated plants, being 21% higher mainly during the 2021/2022 season. However, our results showed that SA application exhibited no changes in ANT in DI plants, in contrast with DI plants not SA-treated during both seasons. Khattab et al. [48] showed that SA application increased the expression of three genes involved in phenol biosynthesis such as *chalcone synthase* genes (*CHS 1*, *CHS 2*, and *CHS 3*), which could be associated with higher phenolic compound levels in our study. However, the molecular mechanism remains unclear [49]. We found higher TSS, TP, AA, and ANT levels in *A. chilensis* fruits as compared to González et al. [47], which could be explained by genetic and environmental factors. In fact, Fredes et al. [50] reported genetic differences associated with ANT accumulation in *A. chilensis* fruits. Our results agree with Giménez et al. [45], who showed that 0.5 mM SA application significantly increased TP (by about 20%) and AA (by about 50%) in fruits of *P. avium*. Recently, Retamal-Salgado et al. [51] showed that 2 mM SA application increased by 100% TP, AA and ANT in *V. corymbosum* fruits. In contrast, we did not observe changes in ANT between SA treatments in DI plants. García-Pastor et al. [46] reported that ANT levels were SA-dose-dependent in *P. granatum* fruits, showing that the highest ANT levels were found under a 10 mM-SA dose. Because yield was reduced by a greater extent than was the reduction in individual fruit weight, it is possible to speculate that fruit drop occurred because of the effect of deficit irrigation and the absence of SA application. Regarding irrigation treatment, several reports indicate that fruit drop can be increased by adverse conditions such as suboptimal irrigation, as confirmed by previous research on *P. granatum* [52], *Prunus armeniaca* [53], and *Prunus salicina* [54]. The efficacy of foliar applications of SA in improving fruit retention has been also reported in *Citrus reculata* [55] and *Mangifera indica* [56], as SA treatment probably inhibits ethylene synthesis and starch–sugar conversion [57].

## 4. Materials and Methods

### 4.1. Experimental Site

The field experiments were conducted at a commercial fruit orchard, “Fundo Ciruelo,” located in Lanco, Los Ríos Region, Chile (39°30′08″ S; 72° 47′59″ W), during two consecutive seasons (2020/2021 and 2021/2022). The plant materials corresponded to three-year-old *A. chilensis* plants, established using a 3.0 × 2.0 m planting design (1666 trees ha^−1^). Rows were laid out in a north–south direction. The climate of the site is a Mediterranean template, with warm summers [58]. The soil at the experimental site is classified as the Lanco series, (Andisol, Typic Durudands) with moderate depth and good permeability [59]. The soil texture analysis showed a silt loam surface (41.2% sand; 42.7% silt; 16.1% clay). Soil nutrient analysis showed a content of N of 5.3 mg kg^−1^, P of 1.2 mg kg^−1^, K of 56.9 mg kg^−1^, Ca of 1.03 cmol+ kg^−1^, Mg of 0.28 cmol+ kg^−1^, Na of 0.17 cmol+ kg^−1^, pH of 5.5, and organic matter of 9.37%. Agronomic practices of the orchard such as pruning and fertilization were performed according to technical recommendations of Plangen Co., Máfil, Chile. Weeds were controlled by mulching and between rows by mechanical mowing.

### 4.2. Treatments and Meteorological Measurements

*A. chilensis* trees grown under field conditions were subjected to two irrigation treatments: (i) application of 100% of reference evapotranspiration (ETo) (full irrigation (FI)), and (ii) application of 60% ETo (deficit irrigation plants (DI)), according to the data in our previous studies [39,40]. The ETo and rainfall were obtained from the La Paz Automatic Weather Station (AWS) located at a site 10 km from the orchard (https://agrometeorologia.cl, accessed on 15 March 2023). The ETo and rainfall data are shown in Figure 1. The different doses of irrigation water were applied using one irrigation line per row with one dripper per plant of 4 L h^−1^ and 2.4 L h^−1^ (Netafim Ltd., Tel Aviv, Israel) for FI and DI, respectively. The irrigation treatments were made from fruit set (November 05) to fruit harvest (January) (as described by Vogel et al. [60]). At fruit color change (80 days after full bloom), a single application of 0.5 mM salicylic acid (SA) (Sigma, St. Louis, MO, USA) was made by spraying homogeneously fruits and leaves of plants under both irrigation treatments. The SA was dissolved in double-distilled water containing 0.05% (*v*/*v*) of Tween 20 as the surfactant wetting agent (+SA). Double-distilled water containing only 0.05% (*v*/*v*) of Tween 20 was used as the control solution (−SA). The SA dose was selected based on the results in our previous study [39]. Foliar SA treatments were performed on the same day, early in the morning, using a backpack spray pump. Plant water status was determined at fruit harvest. When fruits were ripe, leaves and ripe fruits were harvested, and immediately dipped in liquid nitrogen and stored at −80 °C for further biochemical and antioxidant-related parameter determinations. In addition, ripe fruits were harvested, stored in a portable refrigerator, and transferred to the laboratory for fruit quality-related parameter determinations.

### 4.3. Plant Water Status

Stem water potential (Ψ_S_) was measured once at fruit harvest between 08:00 and 10:00 h. For this, leaves were selected from the middle third and sun-exposure part of each tree, selecting mature leaves with no visual symptoms of biotic or abiotic stress. Briefly, leaves were covered with aluminum foil in a plastic bag during the 60 min before measurement. The Ψ_w_ was measured with a Scholander chamber Model 1000 (PMS, Instruments Co., Corvallis, OR, USA) according to the Begg and Turner [61] described method.

### 4.4. ABA Determinations in Leaves

The endogenous ABA of leaf samples was determined by isotope dilution analysis, using a protocol similar to that described by Tillmann et al. [62] for indole-3-acetic acid, using a liquid chromatography–high-resolution mass spectrometry (LC-HR-MS) system (Dionex Ultimate 3000 UHPLC, Q Exactive mass spectrometer, Xcalibur software, version 4.1.31.9, Thermo Scientific, Waltham, MA, USA). Briefly, leaf samples were homogenized in a buffered solvent: 65% isopropanol, 35% 0.2 M imidazole (pH 7.0). Deuterated-ABA ([^2^H_6_]ABA) was used as the internal standard, and added to each sample. Samples were incubated on ice for 60 min to allow for [^2^H_6_]ABA isotopic standard equilibration with the endogenous ABA. [^2^H_6_]ABA was synthesized according to the procedure of Drobrev et al. [63]. Samples were pre-purified using NH_2_ resin solid-phase extraction (SPE) TopTip minicolumns. Finally, samples were injected onto the LC-HR-MS system and analyzed using positive ion mode controlled with Xcalibur software. Endogenous ABA was determined using the isotope dilution equation [64] from the ion abundance at the base peak of each compound: the *m*/*z* 247 for endogenous ABA and the m/z 253 for the [^2^H_6_]ABA. The ABA level was expressed as µg per gram dry weight (µg ABA g^−1^ DW).

### 4.5. Lipid Peroxidation in Leaves

Peroxidation of lipid membranes was determined in leaves of *A. chilensis*. Leaf samples were macerated with a mixture of trichloroacetic acid (TCA) and thiobarbituric acid (TBA), and then centrifuged at 13,000 rpm for 10 min at 4 °C, according to the procedure of Du and Bramalage [65]. The resulting absorbance was measured at 532, 600, and 440 nm using a spectrophotometer (UV/VIS Unico SpectroQuest 2800) to correct for the interference generated by thiobarbituric acid reactive substance (TBARS)–sugars complexes. The TBARS content was expressed as nmol of malondialdehyde (MDA) per gram of fresh weight (nmol MDA g^−1^ FW).

### 4.6. Yield and Fruit Quality

Fruits were harvested from the trees with a coffee harvester (Makita DUX60-EJ400MP, Kunshan, China). Fruit yield was determined using a precision balance (Model BA2204B, Biobase Meihua Trading, Jinan, China). A total of 100 fruits per tree was collected for quality parameter determinations including fruit weight, equatorial and polar diameter, total soluble solids, and titratable acidity. Fresh-fruit weight and dry weight were determined using an analytical balance. For dry weight, fresh fruits were dried in an oven for 72 h at 60 °C. Equatorial and polar diameter of fresh fruits were determined using a digital caliper (accuracy ±0.01 mm). Total soluble solids (TSS) and titratable acidity (TA) were determined in the fruit juice using a thermo-compensated digital refractometer (ATAGO, Mod. PAL-BX I ACID F5, Saitama, Japan) and expressed as °Brix and percentage (%) of citric acid, respectively. It is important to mentioned that fruit firmness was not determined, due to the standard practice of *A. chilensis* fruits as processed in the food and pharmacy industry in Chile [34,38].

### 4.7. Determination of Antioxidant-Related Parameters in Fruits

Antioxidant capacity (AA), total phenols (TP), and total anthocyanins (TA) were determined in fruits of *A. chilensis*. All these biochemical parameters were measured using a spectrophotometer (UV/VIS Unico SpectroQuest 2800). For both AA and TP, fruit samples (0.15 g) were macerated with ethanol (80% *v/v* EtOH/water) and then centrifuged at 13,000 rpm for 10 min at 4 °C. The supernatant was collected and used for both determinations. The AA was determined using the stable free radical 2,2-diphenyl-1-picryl-hydrazyl (DPPH), following the protocol described by Chinnici et al. [66]. The absorbance was measured at 515 nm, and the results were expressed as mg of Trolox equivalents per gram of dry weight (mg TE g^−1^ DW). Total phenols were determined by the Folin–Ciocalteau method [67], measuring the absorbance at 765 nm, using caffeic acid as standard. The results were expressed as mg of caffeic acid equivalents per gram of dry weight (mg CAE g^−1^ DW).

Total anthocyanins were determined following the pH differential method, described by Strack and Wray [68]. Briefly, samples were macerated with 1 mL of acidified ethanol and subjected to shaking in dark conditions overnight at 4 °C. The supernatant was collected, and the absorbance was determined at 530 and 675 nm, spectrophotometrically (UV/VIS Unico SpectroQuest 2800). The results are expressed as mg of cyaniding 3-O-glycoside equivalents per gram of dry weight (mg C3G g^−1^ DW).

### 4.8. Experimental Design and Statistical Analysis

The experiment was performed in a randomized complete block design (RCBD) with nine replicates for each treatment. The experiment contemplated two factors; irrigation treatment (two levels: 100% and 60% ETo) and SA application (two levels: 0 and 0.5 mM SA). The Levene and Kolmogorov–Smirnov test was used for testing homogeneity and normality of variances of the data. Then, data were analyzed using a two-way ANOVA (irrigation treatment and SA application). However, no significant interactions were found between factors, and thus a Student’s *t* test was used to compare the two treatments for each factor. The statistical analyses were performed using Sigma Stat v.2.0 (SPSS, Chicago, IL, USA).

## 5. Conclusions

This study showed that using a DI strategy increased ABA levels and oxidative stress, with a decrease in fruit growth parameters and yield compared to full irrigation in *A. chilensis* plants. However, DI conditions saved water in crop production in the context of water deficit and climate change. SA application recovered all fruit parameters and yield such as fruit fresh weight, fruit dry weight, equatorial diameter, and yield in plants subjected to DI, saving 40% of water. Also, the results showed that SA application reduced ABA levels, which suggested a higher stomata opening, thus possibly improving yield.

On the other hand, SA application demonstrated an improvement in fruit quality and antioxidant-related parameters, with the exception of TA and ANT. Therefore, it seems that SA could be a promising agronomic tool for improving fruit yield, quality and antioxidant properties in maqui fruits, increasing the tolerance of plants to deficit irrigation. However, more biochemical and molecular studies under field conditions are needed to confirm these results.

## Figures and Tables

**Figure 1 plants-12-03279-f001:**
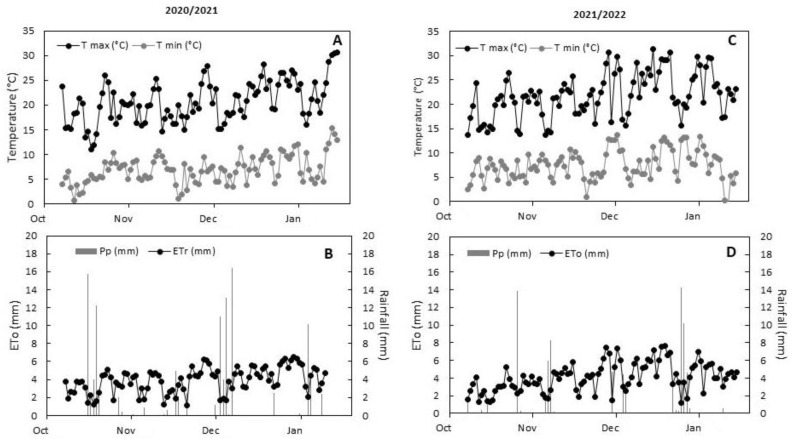
Daily values for maximum (T_max_) and minimum (T_min_) temperature, reference evapotranspiration (ETo), and rainfall during both evaluated seasons. The figures (**A**,**B**) correspond to the 2020/2021 season; the figures (**C**,**D**) correspond to the 2021/2022 season.

**Figure 2 plants-12-03279-f002:**
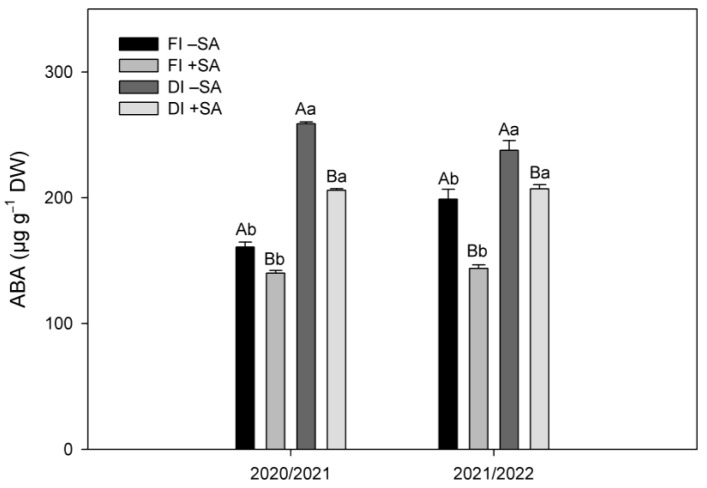
Abscisic acid (ABA) levels in leaves of *A. chilensis* plants subjected to two irrigation treatments (full irrigation (FI) (100% ETo) and deficit irrigation (DI) (60% ETo)) and two SA doses (0 and 0.5 mM) during the two seasons. Different uppercase letters indicate significant differences between SA applications for the same irrigation treatment and season according to Student’s *t* test (*p* ≤ 0.05). Different lowercase letters indicate significant differences between irrigation treatments for the same SA application and season according to Student’s *t* test (*p* ≤ 0.05). The bars are means ± SE (*n* = 9).

**Figure 3 plants-12-03279-f003:**
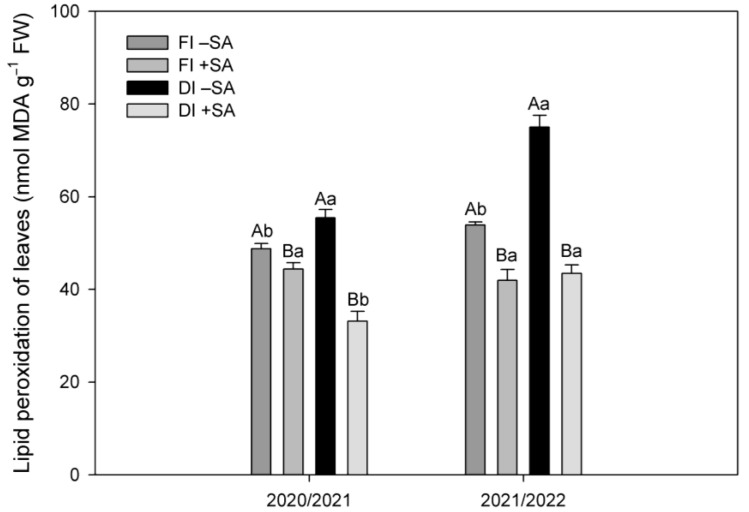
Lipid peroxidation in leaves of *A. chilensis* plants subjected to two irrigation treatments (full irrigation (FI) (100% ETo) and deficit irrigation (DI) (60% ETo)) and two SA doses (0 and 0.5 mM) during two seasons. Different uppercase letters indicate significant differences between SA applications for the same irrigation treatment and season according to Student’s *t* test (*p* ≤ 0.05). Different lowercase letters indicate significant differences between irrigation treatments for the same SA application and season according to Student’s *t* test (*p* ≤ 0.05). The bars are means ± SE (*n* = 9).

**Figure 4 plants-12-03279-f004:**
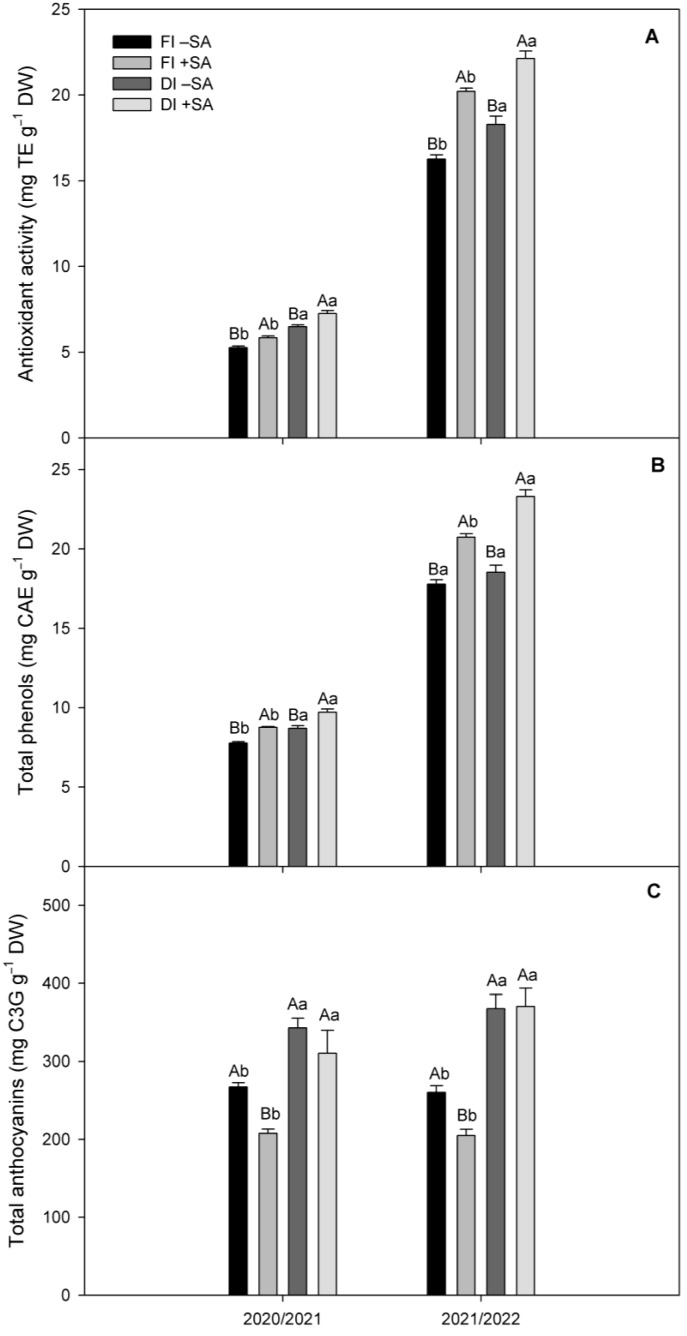
Antioxidant activity (**A**), total phenol content (**B**), and total anthocyanins (**C**) in fruits of *A. chilensis* plants subjected to two irrigation treatments (full irrigation (FI) (100% ETo) and deficit irrigation (DI) (60% ETo)) and two SA doses (0 and 0.5 mM) during two evaluated seasons. Different uppercase letters indicate significant differences between SA applications for the same irrigation treatment according to Student’s *t* test (*p* ≤ 0.05). Different lowercase letters indicate significant differences between irrigation treatments for the same SA application according to Student’s *t* test (*p* ≤ 0.05). The bars are means ± SE (*n* = 9).

**Table 1 plants-12-03279-t001:** Total amount of water applied to *A. chilensis* plants subjected to two irrigation treatments: full irrigation (FI) (100% ETo) and deficit irrigation (DI) (60% ETo) during the two seasons.

Cumulative Values	Season 2020/2021		Season 2020/2021	
	FI	DI	FI	DI
Irrigation (m^3^ ha^−1^)	2717	1630	2991	1795
Rainfall (m^3^ ha^−1^)	115	115	80	80
Total water (m^3^ ha^−1^)	2832	1699	3071	1875

**Table 2 plants-12-03279-t002:** Values of stem water potential (Ψ_w_) measured in *A. chilensis* plants subjected to two irrigation treatments (full irrigation (FI) (100% ETo) and deficit irrigation (DI) (60% ETo)) and two SA doses (0 and 0.5 mM).

Treatment	Season 2020/2021	Season 2020/2021
FI −SA	−0.67 ± 0.02 Aa	−0.65 ± 0.04 Aa
FI +SA	−0.59 ± 0.02 Aa	−0.58 ± 0.03 Aa
DI −SA	−1.31 ± 0.07 Ab	−1.33 ± 0.04 Ab
DI +SA	−1.11 ± 0.07 Bb	−1.17 ± 0.02 Bb

Different uppercase letters indicate significant differences between SA applications for the same irrigation treatment and season according to Student’s *t* test (*p* ≤ 0.05). Different lowercase letters indicate significant differences between irrigation treatments for the same SA application and season according to Student’s *t* test (*p* ≤ 0.05). The data are means ± SE (*n* = 9).

**Table 3 plants-12-03279-t003:** Yield components of *A. chilensis* plants subjected to two irrigation treatments (full irrigation (FI) (100% ETo) and deficit irrigation (DI) (60% ETo)) and two SA doses (0 and 0.5 mM) during two evaluated seasons.

	Fruit Fresh Weight (g of 10 Fruits)	Fruit Dry Weight (g of 10 Fruits)	Yield per Plant (kg)	Total Yield (kg ha^−1^)
**Season 2020/2021**				
FI −SA	2.21 ± 0.02 Aa	0.77 ± 0.04 Aa	2.33 ± 0.2 Aa	3881.7 ± 0.4 Aa
FI +SA	2.35 ± 0.07Aa	0.79 ± 0.07 Aa	2.47 ± 0.3 Aa	4115.0 ± 0.5 Aa
DI −SA	1.53 ± 0.06 Bb	0.54 ± 0.05 Bb	1.75 ± 0.1 Bb	2915.5 ± 0.3 Bb
DI +SA	2.32 ± 0.13 Aa	0.83 ± 0.01 Aa	2.15 ± 0.6 Aa	3581.9 ± 1.0 Aa
**Season 2021/2022**				
FI −SA	2.89 ± 0.53 Aa	0.78 ± 0.02 Aa	2.32 ± 0.2 Aa	3865.1 ± 0.5 Aa
FI +SA	2.84 ± 0.28 Aa	0.81 ± 0.01 Aa	2.51 ± 0.3 Aa	4181.6 ± 0.4 Aa
DI −SA	1.70 ± 0.42 Bb	0.59 ± 0.06 Bb	1.67 ± 0.4 Bb	2782.4 ± 0.6 Bb
DI +SA	3.02 ± 0.34 Aa	0.80 ± 0.05 Aa	2.27 ± 0.6 Aa	3781.8 ± 0.4 Aa

Different uppercase letters indicate significant differences between SA applications for the same irrigation treatment and season according to Student’s *t* test (*p* ≤ 0.05). Different lowercase letters indicate significant differences between irrigation treatments for the same SA application and season according to Student’s *t* test (*p* ≤ 0.05). The data are means ± SE (*n* = 9).

**Table 4 plants-12-03279-t004:** Fruit quality of *A. chilensis* plants subjected to two irrigation treatments (full irrigation (FI) (100% ETo) and deficit irrigation (DI) (60% ETo)) and two SA doses (0 and 0.5 mM) during two evaluated seasons.

	Equatorial Diameter (mm)	Polar Diameter (mm)	Total Soluble Solids (°Brix)	Titratable Acidity (% Citric Acid)
**Season 2020/2021**				
FI −SA	6.75 ± 0.18 Aa	6.83 ± 0.18 Aa	19.20 ± 1.71 Ab	1.22 ± 0.12 Aa
FI +SA	6.76 ± 0.12 Aa	7.07 ± 0.09 Aa	24.57 ± 2.01 Ab	1.18 ± 0.07 Aa
DI −SA	5.87 ± 0.11 Bb	6.33 ± 0.15 Aa	28.23 ± 1.86 Aa	0.81 ± 0.01 Aa
DI +SA	6.71 ± 0.07 Aa	6.98 ± 0.13 Aa	31.23 ± 0.70 Aa	1.02 ± 0.14 Aa
**Season 2021/2022**				
FI −SA	7.26 ± 0.20 Aa	6.85 ± 0.15 Aa	22.53 ± 0.64 Ab	1.20 ± 0.10 Aa
FI +SA	7.41 ± 0.13 Aa	6.88 ± 0.12 Aa	27.07 ± 1.05 Ab	1.07 ± 0.05 Aa
DI −SA	5.82 ± 0.15 Bb	6.44 ± 0.13 Aa	29.05 ± 0.55 Aa	1.14 ± 0.01 Aa
DI +SA	7.53 ± 0.13 Aa	7.04 ± 0.06 Aa	32.47 ± 2.17 Aa	1.23 ± 0.03 Aa

Different uppercase letters indicate significant differences between SA applications for the same irrigation treatment and season according to Student’s *t* test (*p* ≤ 0.05). Different lowercase letters indicate significant differences between irrigation treatments for the same SA application and season according to Student’s *t* test (*p* ≤ 0.05). The data are means ± SE (*n* = 9).

## Data Availability

The data presented in this study are available in the Section 2.

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
