# Peer review of "Pre-Harvest Salicylic Acid Application Affects Fruit Quality and Yield under Deficit Irrigation in Aristotelia chilensis (Mol.) Plants"

_plants, 2023, doi:10.3390/plants12183279_

Round 1

Reviewer 1 Report

The MS “Pre-harvest salicylic acid application affects the ABA levels, antioxidant-related parameters, fruit quality, and yield in Aristotelia chilensis (Mol.) plants under deficit irrigation” has some interests in production, I have some concers need to be addressed.

1.  I suggest to reviest the title as: Pre-harvest salicylic acid application affects fruit quality and yield under deficit irrigation in Aristotelia chilensis (Mol.).

2. Why did you choose 0.5 mM SA, what about other concentrations?

3. if the ABA content in leaf has some relationship with the fruit qualtiy, why did you not measure the physiological index in fruit.

4. The SA applicaion had a big effect on the fruit yield, how to collect the data, please expaint them in detail.

5. What about the sugar content after SA application, if the SA delayed the fruit developement process.

6. Did you measure the endogenous SA in leaf and fruit?

Author Response

Reviewer #1

The MS “Pre-harvest salicylic acid application affects the ABA levels, antioxidant-related parameters, fruit quality, and yield in Aristotelia chilensis (Mol.) plants under deficit irrigation” has some interests in production, I have some concerns need to be addressed.

We thank to reviewer. The corrections have been written with a track in the manuscript. To facilitate your evaluation, the following is a point by point response to the questions and comments.

1.  I suggest to reviest the title as: Pre-harvest salicylic acid application affects fruit quality and yield under deficit irrigation in Aristotelia chilensis (Mol.).

Thank to reviewer. We modified the title. Please see line: 2-5.

2. Why did you choose 0.5 mM SA, what about other concentrations?

We thank to reviewer. We choose 0.5 mM SA doses because we observed that 0.5 mM SA improved physiological performance and antioxidant properties in our previous study (González-Villagra et al. 2022). We are planning a experiment with more SA doses under field conditions during the next season.

3. if the ABA content in leaf has some relationship with the fruit qualtiy, why did you not measure the physiological index in fruit.

Thank to reviewer. We did not associate ABA levels from leaves to ripening of fruits. We did not measure ABA in fruits. We planning measure ABA in fruits in the experiment next season.

4. The SA application had a big effect on the fruit yield, how to collect the data, please expaint them in detail.

Thank to reviewer. We harvested all fruits from the tree with the coffe-harvester. Then, fruits were collected in a plastic box and weighted in the precision balance.

5. What about the sugar content after SA application, if the SA delayed the fruit development process.

We observed that SA increased total soluble solids. However, SA did not modify ripening processes in A. chilensis. All fruits were harvested at dark color.

6. Did you measure the endogenous SA in leaf and fruit?

Thank to reviewer. We did not measure SA in this experiment. Unfortunately, due to we did not have more leaf samples to do it. However, we are planning to measure SA in the next experiment.

*references to lines in the text of our answers are referred to the word version of the revised manuscript.

Reviewer 2 Report

The paper is very well written.  The experiments were repeated and the finding have immediate applicability. 

Minor comments:

Why were the results related to the analyses of antioxidant activities omitted from the abstract ? How did the authors determine 40% on water use?

Please adopt the same principle for the spelling of Latin names, i.e. either give the classifiers or not, I propose to remove the classifiers (lines 57,58, 60).

What was the salicylic acid dissolved in, as it is in powder form?

The discussion could be more in-depth

Author Response

Reviewer #2

The paper is very well written.  The experiments were repeated and the finding have immediate applicability. 

Minor comments:

Thank to reviewer. To facilitate your evaluation, the following is a point by point response to the questions and comments.

1.   Why were the results related to the analyses of antioxidant activities omitted from the abstract?

Thank to reviewer. We included a sentence of antioxidant activities in the abstract. Please see line 39-41 (abstract section).

2.   How did the authors determine 40% on water use?

We subtract total water applied from both irrigation treatment (FI-DI) from the table 1.

3. Please adopt the same principle for the spelling of Latin names, i.e. either give the classifiers or not, I propose to remove the classifiers (lines 57,58, 60).

Thank to reviewer. We deleted the classifiers throughout the manuscript. Please see line 61-64.

4. What was the salicylic acid dissolved in, as it is in powder form?

The SA was dissolved in double-distilled water. Please see line 353.

5. The discussion could be more in-depth

Thank to reviewer. We improved our discussion.

*references to lines in the text of our answers are referred to the word version of the revised manuscript.
